# In Vitro: The Effects of the Anticoccidial Activities of *Calotropis procera* Leaf Extracts on *Eimeria stiedae* Oocysts Isolated from Rabbits

**DOI:** 10.3390/molecules28083352

**Published:** 2023-04-10

**Authors:** Mutee Murshed, Hossam M. A. Aljawdah, Mohammed Mares, Saleh Al-Quraishy

**Affiliations:** Department of Zoology, College of Science, King Saud University, P.O. Box 2455, Riyadh 11451, Saudi Arabia

**Keywords:** *Eimeria stiedae*, *Calotropis procera*, oocysts, inhibition, in vitro, amprolium

## Abstract

Hepatic coccidiosis is an infectious and mortal disease that causes global economic losses in rabbits. The research aimed to assess the efficacy of *Calotropis procure* leaf extracts on the inhibition of *Eimeria stiedae* oocysts and to determine the optimal dosage for suppressing the parasite’s infective phase. In this experiment, oocyst samples per milliliter were tested, and 6-well plates (2 mL) of 2.5% potassium dichromate solution containing 10^2^ non-sporulated oocysts on *Calotropis procera* leaf extracts were exposed after 24, 48, 72, and 96 h, and the treatments were as follows: a nontreated control, 25%, 50%, 100%, and 150% of *C. procera* for oocyst activities. In addition, amprolium was utilized as a reference drug. The *Calotropis procera* was analyzed by GC-Mass, and results showed that the botanical extract contained 9 chemical components that were able to inhibit the oocysts of *E. stiedae* at 100% and 150% concentrations by about 78% and 93%, respectively. In general, an increase in the incubation period and a greater dose resulted in a decrease in the inhibition rate. The results showed that *C. procera* has an effective ability, inhibitory potential, and protective effect on the coccidian oocyst sporulation of *E. stiedae*. It can be used in the disinfection and sterilization of poultry and rabbit houses to get rid of *Eimeria* oocysts.

## 1. Introduction

In recent years, the intensive breeding of rabbits has played an important role in covering people’s needs for meat. Rabbit products are the main diet components, thus making them a suitable and important source of protein. In addition, they are devoid of fat. The effectiveness of obtaining products derived from rabbits is impaired when the animals are infected with diseases, including intestinal parasites such as *Eimeria* spp. [1,2].

*Eimeria stiedae* is the name given to the parasite that is responsible for causing hepatic coccidiosis [3,4]. Since it was found more than a hundred years ago, it has been known to be one of the most common parasites that affect rabbit colonies, which can cause big problems. Its pathogenicity may vary from moderate to severe. It is accompanied by a loss of weight, light intermittent to severe diarrhea, and stools that include mucus or blood, which can lead to dehydration and a reduction in rabbit breeding. There is a possibility of an increase in the death rate [5,6]. Adults are asymptomatic carriers of coccidioidomycosis, and they are a source of acute infection with clinical indications that can lead to infection and the death of young rabbits [7,8,9]. It is one of the most dangerous intracellular and extraintestinal coccidia species that parasitizes liver cells. It belongs to the fourteen different Eimeria species that affect rabbits all over the world [10,11]. Coccidiosis costs rabbit production USD 127 million per year, and comparable losses might occur universally [12]. The appearance of anticoccidial drug resistance catalyzed research for alternative products [13,14].

In order for intensive rabbit farming to work, live vaccinations and anticoccidial drugs put into the feed must be used to prevent coccidiosis. Over the years, drug resistance has made it so that coccidiostats have lost some of their effectiveness. Subclinical coccidiosis is caused by drug-resistant strains of *Eimeria*, high costs, and worries about residues in food and the environment. These are also to blame for the subsequent drop in economic performance, as measured by body weight gain and feed conversion ratio [15,16]. Natural compounds derived from plants were employed by researchers to combat parasites because these products have the potential to be useful sources for innovative anti-parasite candidate drugs. These drugs are not only effective against the parasites, but they may also protect organs in the parasite-infected target hosts [17]. 

*Calotropis procera* leaf extracts (CPLE), which are sometimes called “Madar,” are a medicinal herb that is used in many different herbal mixtures to treat many different illnesses and physical problems. Madar is also a common name for the plant [18]. There have been several studies about the possible remedial qualities of CPLE, including its anti-cancer, anti-radiation, antimicrobial, anti-aging, and anti-stress capabilities. It has been shown that the aqueous extract of the flower has properties that are analgesic, antipyretic, and anti-inflammatory [19,20,21]. It is a well-known tribal plant that is used in a wide variety of traditional medicines to treat a variety of illnesses, including skin conditions, elephantiasis, toothaches, asthma, leprosy, and rheumatism [22]. Another essential component of Ayurvedic medicine is the *C. procera* plant. It is included in the treatment of many different disorders in the form of polyherbal medicines. One of the plants, Ayurveda, has been investigated for its potential anticonvulsant activity [23]. The dried plant may be used as a tonic and has expectorant and anthelminthic properties [24,25].

*Calotropis procera* has phytochemical components such as saponins, alkaloids, tannins, flavonoids, glucosides, and terpenoids. According to several studies, Table 1 shows the chemical compounds available in CPLE shrubs. These components have therapeutic features, including antimicrobial, antioxidant, anti-inflammatory, analgesic, spasmolytic, anti-fertility, and anti-ulcer activity. According to the results, CPLE has a favorable impact, has an inhibitory capacity, and protects against the sporulation of coccidian oocysts and sporozoites when tested in vitro [26]. Recently, many plant extracts, such as garlic [27], *Rumex nervosus*, and *Vitis vinifera*, have been used against *Eimeria* spp. [28,29].

The purpose of this study is to assess the effectiveness of *Calotropis procera* leaves in vitro as a bioindicator of inhibition and the sporulation oocysts of *E. stiedea* isolated from naturally infected rabbits. 

## 2. Results

The botanical chemical analysis obtained by GC-Mass (Figure 1) showed that CPE contained nine chemically active compounds at different peak areas and retention times. 1-Amino-2,6-dimethyl piperidine, 4H-Pyran-4-one, 2,3-dihydro-3,5-dihydroxy-6-methyl-, 7-Ethyl-4-decent-6-one, β-D-Glucopyranose, 1,6-anhydrous-, n-Hexadecanoic acid, Linoleic acid, Oleic Ac-id, Octadecanoic acid, and 9,12-Octadecadienoylchloride, (Z, Z), (Table 2). Each compound and its pharmacological significance against parasites and other diseases were described as shown in (Table 3).

Figure 2A shows the development of oocysts into spores and sporozoites in the control group and Figure 2B shows the deformation and inhibition of oocysts in the treated concentrations-groups.

The effects of sporulation time and experimental groups treated with CPLE on sporulated inhibitory and sporulation of *E. stiedae* oocysts were investigated in vitro. Results showed that as incubation time increased, so did the sporulation percentage, and vice versa for the inhibition percentage. The rate of sporulation inhibition increased significantly with incubation time (*p* ≤ 0.05). 

The proportion of inhibition oocysts was calculated for the control, reference therapy, and experimental groups at different times (24, 48, 72, and 96 h). It is important to note that oocysts began to sporulate after 24 h of incubation. Concentrations (25 and 50 mg/mL) resulted in lower oocyst inhibition percentages. The concentrations of the Madar leaf extracts were inversely proportional to the sporulation inhibition percentage of *E. stiedae* oocysts and directly proportional to the increase in time (Figure 3).

The follow-up to the inhibition of sporulation continued after 24 h of *E. stiedae* oo-cyst incubation in different concentrations with CPLE, and we noticed that sporulation was low in the control group and the concentrations (100, 50, and 25 mg/mL), while at 150 mg/mL and amprolium at 28 mg/mL, there was no sporulation (Figure 4). However, a concentration of 150 mg/mL showed less than 97% inhibitory oocysts. Other concentrations showed different levels of sporulation with time increasing. While the control group (2.5% potassium dichromate solution) remained continuously sporulation throughout the incubation time until 96 h, we note significant differences between treatments with all concentrations of CPLE compared with the reference drug and 2.5% potassium dichromate control group. CPLE (150 mg/mL and amprolium 28 mg/mL) inhibited *E. stiedae* oocysts at marked levels after a 48-h incubation period. At the same time, the control showed high levels of sporulated oocysts, whereas concentrations of 100, 50, and 28 mg/mL of CPLE showed low levels of sporulation (Figure 5).

The results also showed that incubation after 72 h with CPLE at concentrations of 150 mg/mL and 28 mg/mL of amprolium inhibited sporulation by nearly 68–79%, while incubation with CPLE at the same concentration and amprolium for 96 h inhibited sporulation by nearly 89% compared to the control group with potassium dichromate (2.5%), which reached 1% (non-sporulation oocysts). In addition, the increased inhibition of oocysts was at various concentrations throughout 72–96 h for CPL extracts compared with reference therapy (Figure 6).

Figure 6 shows the main effects of sporulation time and experimental groups on the sporulation (%) and sporulation inhibition (%) of *E. stiedae* oocysts. Figure 7, however, shows that the inhibition percentage increased significantly with increasing incubation time of up to 72 h (*p* ≤ 0.05). As a result, there was no discernible difference in the sporulation inhibition rate between doses lasting 72 and 96 h. On sporulation and the suppression of *E. stiedae* oocysts, there were interactions between the experimental groups and the time of sporulation (Figure 7).

The experimental groups generally had a substantial impact on the rates of inhibition (%) and sporulation (%). The greatest inhibition rates and lowest sporulation rates were observed for concentrations of CPLE at 150 and 100 mg/mL (*p* ≤ 0.05). The lowest suppression of sporulation was seen at concentrations of 50 and 25, respectively, of CPLE (*p* ≤ 0.05) (Figure 8).

A cumulative comparison was made between the different times (24, 48, 72, and 96 h) for all concentrations (25, 50, 100, and 150 mg/mL) with the control group and the reference drug that was tested (Table 4). Additionally, a cumulative comparison was made between the inhibited and sporulated oocysts (Table 5).

## 3. Discussion

*Eimeria stiedae*, a pathogenic species that inhabits the liver and intestine, causes significant economic losses in several countries around the world. It is identified in the examined fecal samples containing oocysts. The in vitro anticoccidial properties of *C. procera* extracts were assessed based on the quantity and percentage of *Eimeria* oocysts that were destroyed, in addition to the number of oocysts that were inhibited. As sporozoites from sporulated oocysts that have been ingested are released, they penetrate the hepatic cells, where they quickly proliferate and lead to infections caused by *E. stiedae* before oocysts are generated [12]. The progeny develops outside of the host inside the oocysts, and then those progenies become infectious. Eimeriosis medication resistance has been shown in several investigations [47,48]. Because of this, there is a growing interest in the use of natural products as antiparasitic agents since, in comparison to traditional chemical agents, they are more effective, less toxic, and have fewer negative side effects [17]. 

In recent years, a number of plant extracts have been utilized to prevent and cure infectious diseases in animals and poultry, such as coccidiosis. Several studies have shown that the Madar plant’s methanolic extract possesses anticoccidial effects [49,50]. Our research showed that oocysts and damage in *Eimeria* were inhibited in a dose-dependent manner. The used *C. procera* extracts had more active chemical classes that could predict the inhibition of oocysts. The effect of the CPLE on the oocyst’s inhibition of the parasite *E. stiedae* was tested in vitro with various concentrations (25%, 50%, 100 mg/mL, and 150 mg/mL), compared with control (2.5% potassium dichromate solution) and reference drug (Amprolium), over various time periods (48, 72, and 96 h), which demonstrated the inhibitory potential of CPE on the coccidian oocysts. 

According to our results, concentrations of 100 and 150 mg/mL exhibited activity against oocysts. It is shown to have maximum sporulation inhibition activities and is observed to be more effective against *E. stiedae*. Studies on the components of CPE against *Eimeria* oocysts are very few. To our knowledge, these are among the scarce reports in Saudi Arabia using CP extracts to prevent the coccidiosis oocyst sporulation in the parasite *E. Stiedae*. Similar to the present findings observed in vitro sporulation inhibition with *Psidium guajava* leaf extracts in four *Eimeria* species [51]. Since condensed grape tannins have been shown to inhibit endogenous enzyme activities (such as mannitol-1 phosphate dehydrogenase, mannitol-1 phosphatase, mannitol dehydrogenase, and hexokinase), it is possible that CPLE reduced the rate of sporulation by inhibiting or inactivating the enzymes responsible for the sporulation by inhibiting or inactivating the enzymes responsible for the sporulation process as in helminth eggs [52].

This supports the results of other writers who asserted that some plants, including *Pinus radiata* [53], *Aloe vera*, and *Saccharum officinarum* [54], exhibited anticoccidial characteristics [55]. Moreover, oocysts displayed varying degrees of damage, including damage to their cell walls. In some cases, the oocyst wall almost entirely vanished, leaving just free sporocysts visible [56]. The Madar crude leaf extract may have generated anaerobic conditions that led to these results. This could be a result of the plant extract’s active ingredients. The pathogens are negatively impacted by anaerobic digestion, which is a series of microbial activities that convert the organic material in waste into biogas. They were contrasted with controls, which were different developmental phases of regular *E. stiedae* oocysts. Throughout a 96 h period, a high inhibitory oocyte count was also seen at different dosages of the plant extract when compared to amprolium and the control group.

We figured that the dosage of 150 mg/mL of the extract of CPLE had an effect on the process of oocysts inhibition [57], and this effect was caused by the presence of bioactive phytochemical components. That anti-coccidiosis efficacy demonstrated by CPLE led to a significant elevation in the number of inhibited oocysts from *E. stiedae* oocysts; this corresponds to leaf extracts used from *Vitis vinifera* in vitro, as proven in decreased oocyst output in the feces of mice infected by *Eimeria papillate* [29]. These results are consistent with previous research projects that looked into potential anticoccidial agent sources, such as *Azadirachta indica* [58] *Punica granatum* plant [59], *Ziziphus Spina-Christi* extract [60], mulberry extract [61], and grape [56].

Herbal extracts and other extracts from natural products are often added to poultry feed to help them grow and stay healthy by getting rid of microbes and parasites, especially coccidia (*Eimeria* spp.). In a great number of studies, plant extracts have been shown to improve chicken diets, as well as their performance and the efficiency with which they convert feed to meat [62,63,64]. Also, CPLE may be used as a dietary ingredient in poultry feed because of its anticoccidial activity. Similar in vitro outcomes were discovered by [52], who discovered that an aqueous pine bark extract prevented the sporulation of Eimeria oocysts. The ethnomedicinal herbs’ in vitro anticoccidial effectiveness was shown to be concentration dependent.

Plant extracts with larger dilutions (i.e., 150 and 100 mg/mL) may have higher quantities of the bioactive components, suggesting that the action may be pharmacologically mediated. This corresponds with similar conclusions reached [65,66], and it was also shown that *Thonningia sanguine* and *Aloe vera* extracts had effects on the oocysts of *E. tenella* that depended on concentration [67]. *P. biglobosa* extracts, however, were less effective in preventing the formation of coccidia. This might be due to the absence or presence of only a few amounts of bioactive chemicals. The present analysis discovered that the herbal extracts exhibit anticoccidial activity against *E. stiedae* oocysts, which is contrary to [26], who claimed that there was anticoccidial action shown in vitro of *K. senegalensis*. The appearance of aberrant sporocysts in oocysts subjected to 150 and 100 dilutions of the majority of herbal extracts may be the result of fundamental plant components penetrating the oocysts’ cell walls and damaging the cytoplasm [68], which claimed that there was anticoccidial action shown in vitro of *K. senegalensis*. Furthermore, the appearance of aberrant sporocysts in oocysts subjected to 150 and 100 mg/mL dilutions of the majority of herbal extracts may be the result of fundamental plant components penetrating the oocysts’ cell walls and damaging the cytoplasm [29].

Findings indicate that Madar leaves have a strong effect on the inhibition of *E. stiedae* oocysts and anti-sporulation. It has a potential protective role in the disinfection of poultry and rabbit wards, as well as the sterilization of straws in case of the emergence of infection in chicken and rabbit wards.

## 4. Materials and Methods

### 4.1. Ethical Approval

The research was conducted in accordance with the ethical guidelines for animal use established by the Kingdom of Saudi Arabia (Ethics Committee, King Saud University, Ethics Agreement ID: KSU-SE-21-86).

### 4.2. Preparation of Extract 

The process of preparing the extracts from the *C. procera* leaves in accordance with the technique that was presented by [69]. Fresh leaves and flowers were collected from herbal plant in the desert near Riyadh, Saudi Arabia, and the identity of the plant was validated by a classification scientist at King Saud University’s Botany Department. The leaves were dried for 72 h in an incubator at 45 °C before being ground into a powder and extracted for 24 h at +4 °C by percolating with 80% methanol at room temperature. The extract was then stored for 24 h at 4 °C with intermittent stirring. After this, the extract that was produced was concentrated and dried using a rotating vacuum evaporator (Yamato RE300, Tokyo, Japan) at 40 °C and under decreased pressure. When the crude extracts were produced, they were weighed and then kept at −20 °C until they were employed in an experiment. In order to dissolve the powder for the various experiments, a solution of potassium dichromate (K_2_Cr_2_O_7_) and distilled water was used.

### 4.3. Analysis of Phytochemical Compounds Extracted from C. procera Leaves Using GC-MS

Trace GC-ISQ Quantum mass spectrometer system (Thermo Scientific, Austin, TX, USA) was used to analyze the extract of *C. procera*. The used flow rate was 1 mL.mn^−1^/min. About 1 μL from the sample was injected into a GC-MS equipped with a TG–5MS column (30 m × 0.25 mm ID, 0.25 μm film thickness). Helium gas was used as a carrier at a constant flow of 1.0 mL min^−1^. The mass spectra were observed between 50–500 *m*/*z*. The temperature initially started at 50 °C for 10 min, then increased at a rate of 5 °C min^−1^ to 250 °C, and isothermal was held at 300 °C for 2 min, and finally, isothermal was held for 10 min at 350 °C. NIST, Adams, Terpenoids, and Volatile Organic Compounds libraries were used to identify the phytochemical constituents by comparing the recorded mass spectra for each compound with the data stored in the previous libraries. The relative percentage amount for each component was calculated using Retention time index and comparing its average peak area with total peak areas [70].

### 4.4. Oocyst Sporulation 

Oocysts of *E. stiedae* were isolated from feces and gallbladders of rabbits that had spontaneously contracted the infection. Flotation was used as a means of concentrating the collected oocysts after they had been collected [71]. Oocyst sporulation was carried out in Petri dishes in a moist chamber with temperatures ranging from 25–28 °C, and a humidity of approximately 65% (ETI 6102 Therma-Hygrometer-Indian) in a solution of 2.5% potassium dichromate for 7 days until sporulation. Then, oocyst was washed to get rid of dichromats and pass them via young rabbits. The rabbit is followed for 5–10 days. Fresh feces samples were collected, and they were washed and concentrated according to the method [29], and then they were tested with the extracts (CPLE). 

### 4.5. In Vitro Effect of Extracts on Inhibition of Sporulated Oocyst

A suspension of 800 µL of freshly sporulated oocysts of *E. stiedea* containing 1 × 10^3^ oocysts was added to each 24-well plate, and then these plates were incubated in 2 mL of potassium dichromate with 1 of the following concentrations of CPLE: (25%, 50%, 100%, and 150%). Only the negative control group made use of potassium dichromate solution (K_2_Cr_2_O_7_), whilst the positive control group made use of amprolium 60% (Veterinary and Agricultural Products Company (VAPCO, Amman, Jordan) (EMEA, 2001). We used three different replicates for each different concentration. All 24-well plates for all of the different groups were heated between 25–29 °C, the relative humidity was between 65–75%, and the plates were shaken periodically while they were semi-covered [30]. After 24, 48, 72, and 96 h, a microscopically inspected 10 µL suspension was taken from each group and seen via a light microscope (BX51TF, OLYMPUS, Tokyo, Japan) at a magnification of 40×. Using a light microscope equipped with a McMaster chamber, the viability (oocysts with sporocysts, deformed walls, and inhibitory oocysts) was analyzed, and the percentages of sporocysts and inhibitory oocysts were determined with the use of the following formulae:Sporulation (%) = a number of sporulation oocysts/total number of oocysts × 100.(1)

Inhibition of sporulation (SI%) = Sporulation of control—sporulation of treated/sporulation of control × 100, as described [72].

### 4.6. Statistical Analysis

The ANOVA analysis was performed in one direction, and statistical comparisons between the groups then are made using the Duncan technique. At a significance level of ≤0.05, values have been reported as the mean minus the standard deviation. For the purpose of statistical analysis, the Sigma Plot program, version 11, was applied.

## 5. Conclusions

Findings showed that Madar leaves have a strong effect on the inhibition of *E. stiedae* oocysts and anti-sporulation in vitro. More studies are needed to accurately define the active chemical compounds in CPLE and their modes of action and applications to determine plant extracts that may be an option for the formulation of new products as alternative therapies and determine whether they can be used for in vivo treatment and to understand the histological and molecular mechanisms of CPLE and its protective effects against *E. steidae*-induced hepatic damage. In addition, it can be used to disinfect the floors of poultry and rabbits and fight the fallen oocysts on the floor straw.

## Figures and Tables

**Figure 1 molecules-28-03352-f001:**
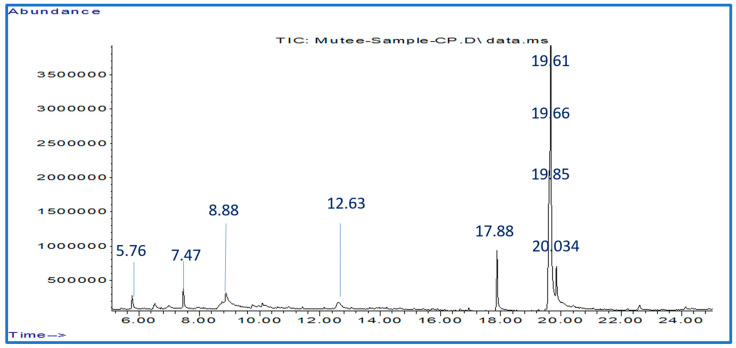
GC–MS chromatogram for the botanical chemical components found in the leaf extracts of *Calotropis procera*.

**Figure 2 molecules-28-03352-f002:**
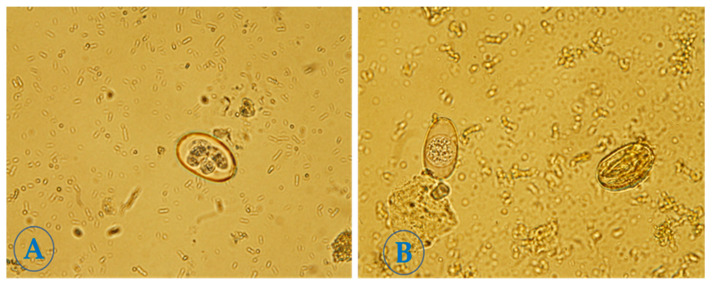
(**A**) Control (2.5% potassium dichromate solution). (**B**) Oocyst treated by *Calotropis procera*.

**Figure 3 molecules-28-03352-f003:**
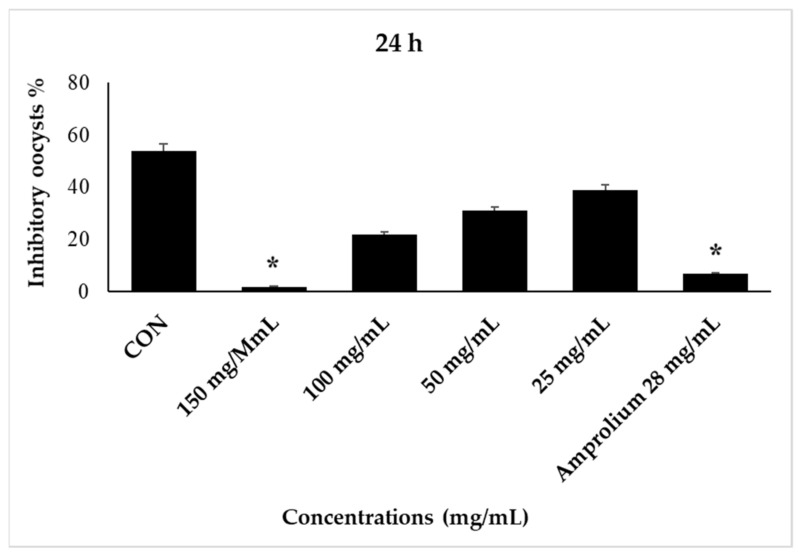
The effect of CP LE on the inhibition of *E. stiedae* oocysts in vitro after 48 h. The significance was assessed by comparing it to a negative control of 2.5% potassium dichromate and a positive control of 28 mg/mL of amprolium. Statistical significance in comparison to the control group (*p* ≤ 0.05). Significance (*): *p*-value ≤ 0.05.

**Figure 4 molecules-28-03352-f004:**
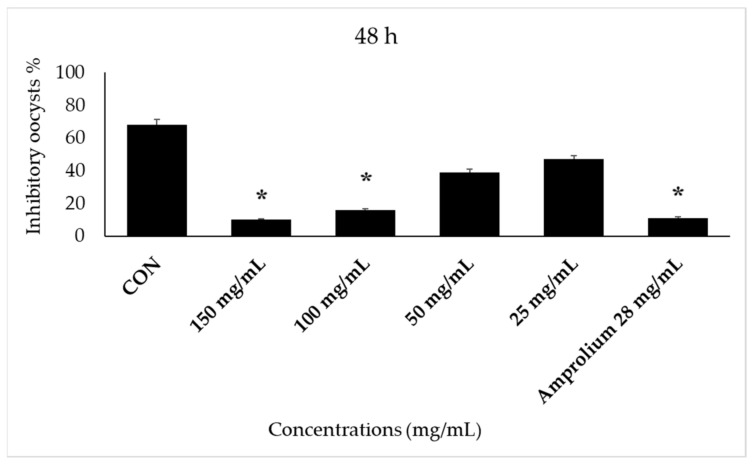
The effect of CP LE on the inhibition of *E. stiedae* oocysts in vitro after 96 h. The significance was assessed by comparing it to a negative control of 2.5% potassium dichromate and a positive control of 28 mg/mL of amprolium. When compared to the control group, there was a significant difference (*p* ≤ 0.05). Significance (*): *p*-value ≤ 0.05.

**Figure 5 molecules-28-03352-f005:**
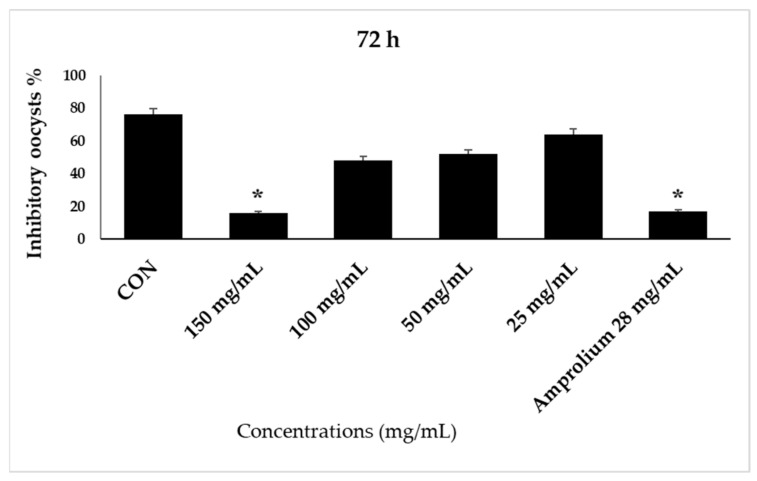
The effect of CP LE on the inhibition of *E. stiedae* oocysts in vitro after 72 h. The significance was assessed by comparing it to a negative control of 2.5% potassium dichromate and a positive control of 28 mg/mL of amprolium. Statistical significance in comparison to the control group (*p* ≤ 0.05). Significance (*): *p*-value ≤ 0.05.

**Figure 6 molecules-28-03352-f006:**
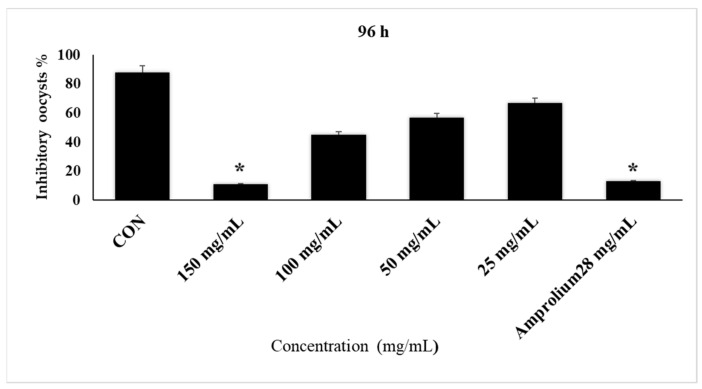
The effect of CPLE on the inhibition of *E. stiedae* oocysts in vitro after 96 h. The significance was assessed by comparing it to a negative control of 2.5% potassium dichromate and a positive control of 28 mg/mL of amprolium. When compared to the control group, there was a significant difference (*p* ≤ 0.05). (h): hours. Significance (*): *p*-value ≤ 0.05.

**Figure 7 molecules-28-03352-f007:**
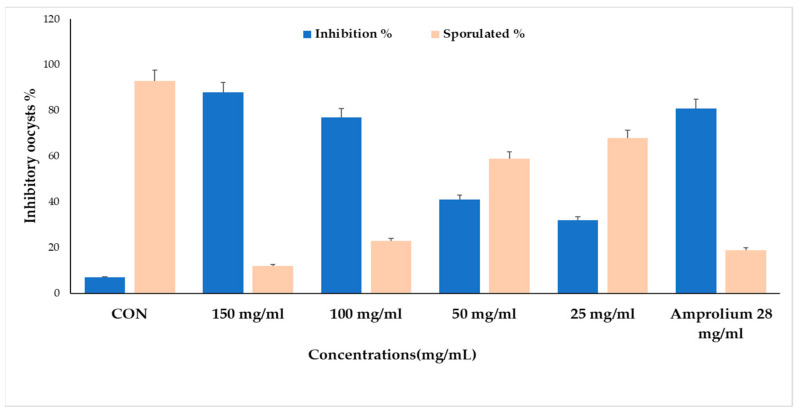
CPLE’s in vitro major contact time effects on inhibition percentage and sporulation percentage of *E. stiedae* oocysts with varied dosages (25, 50, 100, and 150). The significance was assessed by comparing it to a negative control of 2.5% potassium dichromate and a positive control of 28 mg/mL of amprolium.

**Figure 8 molecules-28-03352-f008:**
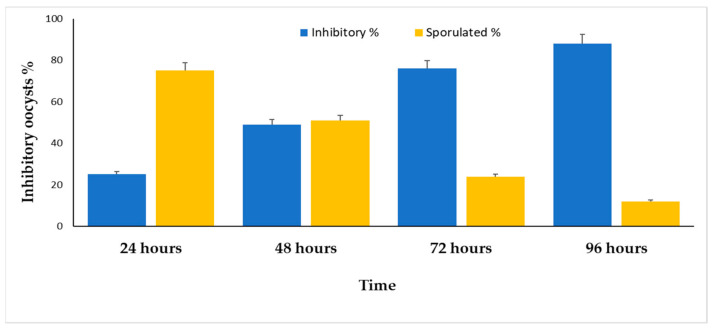
Main effects of CPLE on inhibition of *E. stiedae* oocysts in vitro at varied dosages during 24, 48, 72, and 96 h. The significance was compared to a negative control of 2.5% potassium dichromate and a positive control of 28 mg/mL of amprolium.

**Table 1 molecules-28-03352-t001:** Characterization list of chemical constituents available in the CPLE.

Part of the Plant	Chemical Class	Chemical Ingredient	Reference
Leaves	Flavonoid	α-Amyrin	[30]
		α-Amyrin acetate	[31]
		β-sitosterol	[32]
		Ursolic acid	[33]
		Calotropin	[34]
		Calotropagenin	[34]
		Procesterol (A new steroidal hydroxyl ketone)	[35]
	Other constituents	Calotropenylacetate	[36]
		Multiflorenol	[37]
		16-α-Hydroxy calotropagenin	[38]
		O-Pyrocatechuic acid	[39]
		Quercetin-3-rutinoside	[40]
		α- and β-calotropeols	[40]
		3-epimoretenol	[40]
		Procerain	[40]
		Histamine	[40]

**Table 2 molecules-28-03352-t002:** Identification of phytochemical compounds by GC-Mass in *Calotropis procera* leaves extracts.

NO	RT (min)	Bioactive Phytochemicals Compounds	Molecular Formula	[M − H] (*m*/*z*)	Peak Area %
1	5.76	1-Amino-2,6-dimethylpiperidine	C_7_H_16_N_2_	128	2.09
2	7.47	4H-Pyran-4-one, 2,3-dihydro-3,5-dihydroxy-6-methyl-	C_6_H_8_O_4_	144	2.82
3	8.88	7-Ethyl-4-decen-6-one	C_12_H_22_O	182	10.7
4	12.63	β-D-Glucopyranose, 1,6-anhydro-	C_6_H_10_O_5_	162	4.5
5	17.88	n-Hexadecanoic acid	C_16_H_32_O_2_	256	9.39
6	19.61	Linoleic acid	C_18_H_32_O_2_	280	18.44
7	19.66	Oleic Acid	C_18_H_34_O_2_	282	47.39
8	19.85	Octadecanoic acid	C_18_H_36_O_2_	284	3.43
9	22.61	9,12-Octadecadienoyl chloride, (Z, Z)-	C_18_H_31_ClO	298	1.24

TR: Time retention, M-H: Protonated molecule, MS: Mass acquired range.

**Table 3 molecules-28-03352-t003:** Identification of phytochemical compounds by GC-Mass in ECE-M.

Compound Name	Compound Nature	Function	Reference
1-Amino-2,6-dimethylpiperidine	Nature	Cardiotoxicity Antioxidants Anti-inflammatory arthritis, and muscle and joint pain Antimicrobial	[41]
4H-Pyran-4-one, 2,3-dihydro-3,5-dihydroxy-6-methyl-	Alkaloid	Used for gastrointestinal disorders, gonorrhea, menorrhagia. Antifungal, anti-cancer, sclerosis, and warts	[41]
7-Ethyl-4-decen-6-one	Flavonoid	Antimalarial activity Antifungal Antibacterial Antidiarrhoeal activity	[38]
β-D-Glucopyranose, 1,6-anhydro	Sugar moiety	Antiviral, antifungal Anthelmintic activity Antidiarrhoeal activity Anticonvulsant activity	[38]
n-Hexadecanoic acid	Palmitic acid	Antibacterial, antifungal Antibiofilms Antioxidant and anti-cancer	[42]
Linoleic acid	Linoleic acid	Antibacterial Anti-tumor, anti-virus, and anti-inflammatory	[43]
Oleic Acid	Palmitic acid	Antibacterial, antifungal Antibiofilms Antioxidant and anti-cancer	[44]
Octadecanoic acid	Linoleic acid	Antibacterial, antifungal Antibiofilms Antioxidant and anticancer	[45]
9,12-Octadecadienoyl chloride, (Z, Z)-	solely chloride	Antisecretory, antispermigenic, antitonsilitic, antitubercular, choleretic, contraceptive	[46]

**Table 4 molecules-28-03352-t004:** Cumulative comparison between the different time tests.

	24 h		48 h		72 h		96 h	
	Inhibition	Sporulation	Inhibition	Sporulation	Inhibition	Sporulation	Inhibition	Sporulation
CON	46	54	32	68	24	76	7	93
150 mg/mL	97	3	92	8	88	12	96	4
100 mg/mL	78	22	78	22	52	48	55	45
50 mg/mL	69	31	61	39	48	52	43	57
25 mg/mL	61	39	53	47	26	64	33	67
Amprolium 28 mg/mL	92	8	96	4	97	3	97	3

**Table 5 molecules-28-03352-t005:** Cumulative comparison between the inhibition % and sporulation % test.

	CON	150 mg/mL	100 mg/mL	50 mg/mL	25 mg/mL	Amprolium 28 mg/mL
Inhibition%	11	93	77	41	32	83
Sporulation%	89	7	23	59	68	17

## Data Availability

Not applicable.

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
