# Peer review of "In Vitro: The Effects of the Anticoccidial Activities of Calotropis procera Leaf Extracts on Eimeria stiedae Oocysts Isolated from Rabbits"

_molecules, 2023, doi:10.3390/molecules28083352_

Round 1
Reviewer 1 Report
In the manuscript submitted by Murshed and colleagues entitled "In vitro: Assessing the anticoccidial efficacy of Calotropis procera leaves extracts against Eimeria Stiedae oocysts isolated from naturally infected rabbits", several aspects should be considered:
- I recommend authors follow the guidelines for authors of the respective journal.
- The title of the manuscript should be revised, it is too long, not objective, and appealing.
- The abstract should be rewritten, since it is quite confusing, the information is not properly linked and the main objective of the work is unclear. According to the Molecules guidelines, the abstract should be divided, but without headings, into background, methods, results, and conclusion. The limit of 200 words should not be exceeded. The use of abbreviations should be avoided.
- They should pay attention throughout the document to the way they write. Standardize words in italics. For example, "In vitro", and "m/z" should be in italics, units of measurement (mL not ml). You should have more scientific rigor.
Results
- Figures have poor quality, being inertrepatible. Do Figure 2 referenced in the text before figure 1? Figure 7? Figures should be referenced in order and should be placed as close as possible to their reference in the text.
- You mention statistical treatment, but I can't find it in the results. Check it out.
- Check the references. Table 2.
- The results can be grouped to better follow the values obtained and allow comparison between the different times tested. Could you not summarize them in a table?
Materials and methods
- Shouldn't this be section 4? Check it out.
- Also pay attention to how they reference previous work. For example, in line 328. You should check the guidelines for the correct way to do this.
- Line 331: why 72h? Was it long enough? Or were they based on some literature?
- Why did you use methanol as an extraction solvent? The associated toxic effects are more than known.
- Line 333: I don't understand. Do you percolate for 24 hours, leave the extract for 24 hours in storage, and then evaporate? Confusing. Why do you do that?
- The data on the equipment used should be complete, including information on the molecule, city, and country of production.
- Sections 2.3.1. and 2.3.2. are not being repetitive?
- Did you not use any commercial standards for the characterization of the compounds?
- Was the humidity in the in vitro assays obtained through any equipment? You have to be more specific.
- Why did you choose to do a one-way ANOVA?
References
You should choose to reference more recent literature, preferably from the last decade.
Author Response
Dear Editor-in-Chief
Dear Reviewer
Thank you for accepting the review in evaluating my manuscript entitled (In vitro: Assessing Calotropis procera leaf extracts against Eimeria Stiedae oocysts isolated from rabbits infected).
I answered all the questions and mandatory requirements in detail step by step.
Comments and Suggestions for Authors
Reviewer 1:
Comments and Suggestions for Authors
In the manuscript submitted by Murshed and colleagues entitled "In vitro: Assessing the anticoccidial efficacy of Calotropis procera leaves extracts against Eimeria Stiedae oocysts isolated from naturally infected rabbits", several aspects should be considered:
- I recommend authors follow the guidelines for authors of the respective journal.
Done
- The title of the manuscript should be revised, it is too long, not objective, and appealing.
Done
In vitro: Assessing Calotropis procera leaf extracts against Eimeria Stiedae oocysts isolated from rabbits infected
- The abstract should be rewritten, since it is quite confusing, the information is not properly linked and the main objective of the work is unclear. According to the Molecules guidelines, the abstract should be divided, but without headings, into background, methods, results, and conclusion. The limit of 200 words should not be exceeded. The use of abbreviations should be avoided.
Done: 211 words, with the use of abbreviations be avoided
- They should pay attention throughout the document to the way they write. Standardize words in italics. For example, "In vitro", and "m/z" should be in italics, units of measurement (mL not ml). You should have more scientific rigor.
Done in all the manuscript
Results
- Figures have poor quality, being inertrepatible. Do Figure 2 referenced in the text before figure 1? Figure 7? Figures should be referenced in order and should be placed as close as possible to their reference in the text.
Done
- You mention statistical treatment, but I can't find it in the results. Check it out.
Done
- Check the references. Table 2.
Done
- The results be grouped to better follow the values obtained and allow
Cumulative comparison between the different times tested. Could you not summarize them in a table?
Figure 3. Cumulative comparison between the different times test
|
|
24 h |
48 h |
72 h |
96 |
||||
|
Inhibition |
Sporulation |
Inhibition |
Sporulation |
Inhibition |
Sporulation |
Inhibition |
Sporulation |
|
|
CON |
46 |
54 |
32 |
68 |
24 |
76 |
7 |
93 |
|
150 mg/mL |
97 |
3 |
92 |
8 |
88 |
12 |
91 |
9 |
|
100 mg/mL |
78 |
22 |
78 |
22 |
52 |
48 |
55 |
45 |
|
50 mg/mL |
69 |
31 |
61 |
39 |
48 |
52 |
43 |
57 |
|
25 mg/mL |
61 |
39 |
53 |
47 |
26 |
64 |
33 |
67 |
|
Amprolium 28 mg/mL |
92 |
8 |
96 |
4 |
97 |
3 |
97 |
3 |
Cumulative comparison between the Inhibition% and Sporulation% test
|
CON |
150 mg/mL |
100 mg/mL |
50 mg/mL |
25 mg/mL |
Amprolium 28 mg/mL |
|
|
Inhibition% |
11 |
93 |
77 |
41 |
32 |
83 |
|
Sporulation% |
89 |
7 |
23 |
59 |
68 |
17 |
Materials and methods
- Shouldn't this be section 4? Check it out.
Done
- Also pay attention to how they reference previous work. For example, in line 328. You should check the guidelines for the correct way to do this.
Done
- Line 331: why 72h? Was it long enough? Or were they based on some literature?
72 hours was about enough time for the leaves to dry completely and prepare for grinding.
- Why did you use methanol as an extraction solvent? The associated toxic effects are more than known.
Methanol is the most commonly used extraction solvent due to its high polarity which could produce high extraction yields. According to most studies extracted with methanol.
- Line 333: I don't understand. Do you percolate for 24 hours, leave the extract for 24 hours in storage, and then evaporate? Confusing. Why do you do that?
- procera Leaves extract was prepared according to the method described by Manikandan et al. (2008) with some modification
- The data on the equipment used should be complete, including information on the molecule, city, and country of production.
- Sections 2.3.1. and 2.3.2. are not being repetitive?
Done
- Did you not use any commercial standards for the characterization of the compounds?
Done
- Did you not use any commercial standards for the characterization of the compounds?
Done
- Was the humidity in the in vitro assays obtained through any equipment? You have to be more specific.
(ETI 6102 Therma-Hygrometer- Indian)
The 6102 thermal-hygrometer
- Why did you choose to do a one-way ANOVA?
We use a one-way ANOVA because we have data on one category independent variable and one quantitative dependent variable.
References
You should choose to reference more recent literature, preferably from the last decade.
Done

Reviewer 2 Report
In the present study, authors assessed the anticoccidial efficacy of Calotropis 2 procera leaves extracts against Eimeria Stiedae oocysts isolated 3 from rabbits infected naturally. This research tends to analyze the application value of natural products. However, the following aspects need to be improved in this manuscript:
1. Which ingredient in the extract is the main ingredient responsible for against Eimeria Stiedae oocysts?
2. The extract has significant biological effects only at the concentration of 150mg/ml, which is much higher than the concentration of the positive control drug? Its physiological significance needs to be explained.
3. The author mentioned that natural products are safer. What is the lethal concentration of this extract in mice? Is it safe?
4. Table 2 is more like writing a review.
Author Response
Dear Editor-in-Chief
Dear Reviewer
Thank you for accepting the review in evaluating my manuscript entitled (In vitro: Assessing Calotropis procera leaf extracts against Ei-meria Stiedae oocysts isolated from rabbits infected).
I answered all the questions and mandatory requirements in detail step by step.
Comments and Suggestions for Authors
Reviewer 2:
Comments and Suggestions for Authors
In the present study, authors assessed the anticoccidial efficacy of Calotropis 2 procera leaves extracts against Eimeria Stiedae oocysts isolated 3 from rabbits infected naturally. This research tends to analyze the application value of natural products. However, the following aspects need to be improved in this manuscript:
- Which ingredient in the extract is the main ingredient responsible for against Eimeria Stiedae oocysts?
The most active phytochemical compounds were determined in Calotropis procera leaf extract by GC-Mass through Peak area %:
Oleic Acid = 47.39%,
Linoleic acid=18.44%,
7-Ethyl-4-decent-6-one=10.7%
n-Hexadecanoic acid=9.39%
|
NO |
RT (min) |
Bioactive phytochemicals compounds |
Molecular Formula |
[M-H] (m/z) |
Peak area % |
|
1 |
5.76 |
1-Amino-2,6-dimethylpiperidine |
C7H16N2 |
128 |
2.09 |
|
2 |
7.47 |
4H-Pyran-4-one, 2,3-dihydro-3,5-dihydroxy-6-methyl- |
C6H8O4 |
144 |
2.82 |
|
3 |
8.88 |
7-Ethyl-4-decen-6-one |
C12H22O |
182 |
10.7 |
|
4 |
12.63 |
β-D-Glucopyranose, 1,6-anhydro- |
C6H10O5 |
162 |
4.5 |
|
5 |
17.88 |
n-Hexadecanoic acid |
C16H32O2 |
256 |
9.39 |
|
6 |
19.61 |
Linoleic acid |
C18H32O2 |
280 |
18.44 |
|
7 |
19.66 |
Oleic Acid |
C18H34O2 |
282 |
47.39 |
|
8 |
19.85 |
Octadecanoic acid |
C18H36O2 |
284 |
3.43 |
|
9 |
22.61 |
9,12-Octadecadienoyl chloride, (Z, Z)- |
C18H31ClO |
298 |
1.24 |
- The extract has significant biological effects only at the concentration of 150mg/ml, which is much higher than the concentration of the positive control drug? Its physiological significance needs to be explained.
This is due to the fact that the extract contains many compounds that vary in the percentage of their activity, which shows the effectiveness more. While the reference drug is manufactured from certain compounds with specific proportions of active compounds
- The author mentioned that natural products are safer. What is the lethal concentration of this extract in mice? Is it safe?
Yes, natural products have proven to be safer by not acquiring the parasite's immunity against commercial drugs, in addition to secondary and cumulative effects.
Through previous studies on mice, the highest concentration is non-lethal for mice and effective against the parasite at approximately 200 mg/kg. while more will be fatal to mice.
- Table 2 is more like writing a review.
Done

Author Response
Dear Editor-in-Chief
Dear Reviewer
Thank you for accepting the review in evaluating my manuscript entitled (In vitro: Assessing Calotropis procera leaf extracts against Ei-meria Stiedae oocysts isolated from rabbits infected).
I answered all the questions and mandatory requirements in detail step by step.
Comments and Suggestions for Authors
Reviewer 3:
Abstract:
- Improve English
Done
- Although the results have been described very well but it can be improved
Done
- Conclusion part can be improved
Done
- Check keywords
Main body:
- Similarity index of this article is very high, it should be less than 20%
Done
- English can be improved
Done
- Line 40-42 missing reference
Done
- Authors should compare some other botanical plant leaves with the current study
to support why the extract of Calotropis procera leaves are better, you can add
some previous updated references.
Done
- Chemical composition of Calotropis procera leaves should be discussed here.
Done
- Briefly explain the purpose of this study at the end of the introduction.
Done
- Avoid to use every line as a paragraph as it distracts the readability, try to
summarize small lines into one paragraph in results section.
Done
- Have you placed any results related to the mass spectroscopy in the material and
methods or results? if no then why?
Done
- Discussion portion can be improved
Done
- Conclusion can be more accurately explained
Done

Reviewer 4 Report
In general the study focus on how well Calotropis procera plants work as a biomarker against sporulation and the shape of E. stiedea oocysts that were taken from rabbits that had the parasite naturally. The paper could be considered to be accepted with some suggestion of improvement as follows:
Abstract:
Justification on important of the efficacies of Calotropis procure leaf extracts (CPLE) on the inhibition and sporulation of Eimeria stiedae oocysts should be provided
Brief conclusion of the study should be supplied at the end of abstract part
Introduction:
Details literature supported with related and recent references on the following topics should be provided:
· Properties of aqueous extract from different species of flowers
· Different natural compounds derived from plants and their ability to combat parasites
Results and discussion:
More details technical discussion supported with related and recent references should be provided for the following topics:
· the development of oocysts into spores and sporozoites in the control group
· Effect of concentration on the oocyst inhibition percentages
· Inhibition of sporulation of E. stiedae oocyst at different concentrations with CPLE
· Addition of the Herbal extracts and other extracts from natural products on the grow and stay healthy by getting rid of microbes and parasites
· Mechanism on the anti-coccidial action after the herbal extracts addition
Author Response
Dear Editor-in-Chief
Dear Reviewer
Thank you for accepting the review in evaluating my manuscript entitled (In vitro: Assessing Calotropis procera leaf extracts against Ei-meria Stiedae oocysts isolated from rabbits infected).
I answered all the questions and mandatory requirements in detail step by step.
Comments and Suggestions for Authors
Reviewer 4:
Comments and Suggestions for Authors
In general, the study focus on how well Calotropis procera plants work as a biomarker against sporulation and the shape of E. stiedea oocysts that were taken from rabbits that had the parasite naturally. The paper could be considered to be accepted with some suggestion of improvement as follows:
Abstract:
Justification on important of the efficacies of Calotropis procure leaf extracts (CPLE) on the inhibition and sporulation of Eimeria stiedae oocysts should be provided
Done
Brief conclusion of the study should be supplied at the end of abstract part
Done
Introduction:
Details of literature supported with related and recent references on the following topics should be provided:
- Properties of aqueous extract from different species of flowers
Done
- Different natural compounds derived from plants and their ability to combat parasites
Done
Results and discussion:
More details technical discussion supported with related and recent references should be provided for the following topics:
- the development of oocysts into spores and sporozoites in the control group
Done
- Effect of concentration on the oocyst inhibition percentages
Done
- Inhibition of sporulation of E. stiedae oocyst at different concentrations with CPLE
Done
- Addition of the Herbal extracts and other extracts from natural products on the grow and stay healthy by getting rid of microbes and parasites
Done
- Mechanism on the anti-coccidial action after the herbal extracts addition
Done

Round 2
Reviewer 1 Report
I recommend the authors take into consideration the comments of my first review. Which are still to be applied to this reviewed version.
I consider that the manuscript doesn't fit the scope of Molecules and doesn't have the scientific quality to be considered for publication.
Author Response
Dear Reviewer
Thank you for accepting the review in evaluating my manuscript entitled (In Vitro: The Effects of the Anticoccidial Activities of Calotropis procera Leaf Extracts on Eimeria Stiedae Oocysts Isolated of Rabbits).
I answered all the questions and mandatory requirements in detail step by step marked in light blue
I answered all the questions and mandatory requirements in detail, step by step, marked in light blue.
Comments and Suggestions for Authors
I recommend the authors take into consideration the comments of my first review. Which are still to be applied to this reviewed version.
The first review was reconsidered, confirming the comments as correct, as follows:
- I recommend authors follow the guidelines for authors of the respective journal.
The guidelines of the journal's authors were followed from the introduction to the references, including formats, figures, tables, and others
- The title of the manuscript should be revised, it is too long, not objective, and appealing.
Done, Changed the intitle of (In vitro: Assessing Calotropis procera leaf extracts against Eimeria Stiedae oocysts isolated from rabbits infected)
to (TIn Vitro: The Effects of the Anticoccidial Activities of Calotropis procera Leaf Extracts on Eimeria Stiedae Oocysts Isolated of Rabbits)
- The abstract should be rewritten, since it is quite confusing, the information is not properly linked and the main objective of the work is unclear. According to the Molecules guidelines, the abstract should be divided, but without headings, into background, methods, results, and conclusion. The limit of 200 words should not be exceeded. The use of abbreviations should be avoided.
Done: 211 words, with the use of abbreviations be avoided
The abstract has been rewritten as requested by the reviewer.
- They should pay attention throughout the document to the way they write. Standardize words in italics. For example, "In vitro", and "m/z" should be in italics, units of measurement (mL not ml). You should have more scientific rigor.
Has been consolidated standardize words in italics in throughout the document
Results
- Figures have poor quality, being inertrepatible. Do Figure 2 referenced in the text before figure 1? Figure 7? Figures should be referenced in order and should be placed as close as possible to their reference in the text.
Done, the numbers are indicated in order.
- You mention statistical treatment, but I can't find it in the results. Check it out.
We use a one-way ANOVA because we have data on one category independent variable and one quantitative dependent variable.
- Check the references. Table 2.
Complete table references have been added
- The results be grouped to better follow the values obtained and allow
Cumulative comparison between the different times tested. Could you not summarize them in a table?
They are summarized in a table
Figure 3. Cumulative comparison between the different times test
|
|
24 h |
48 h |
72 h |
96 |
||||
|
Inhibition |
Sporulation |
Inhibition |
Sporulation |
Inhibition |
Sporulation |
Inhibition |
Sporulation |
|
|
CON |
46 |
54 |
32 |
68 |
24 |
76 |
7 |
93 |
|
150 mg/mL |
97 |
3 |
92 |
8 |
88 |
12 |
91 |
9 |
|
100 mg/mL |
78 |
22 |
78 |
22 |
52 |
48 |
55 |
45 |
|
50 mg/mL |
69 |
31 |
61 |
39 |
48 |
52 |
43 |
57 |
|
25 mg/mL |
61 |
39 |
53 |
47 |
26 |
64 |
33 |
67 |
|
Amprolium 28 mg/mL |
92 |
8 |
96 |
4 |
97 |
3 |
97 |
3 |
Cumulative comparison between the Inhibition% and Sporulation% test
|
CON |
150 mg/mL |
100 mg/mL |
50 mg/mL |
25 mg/mL |
Amprolium 28 mg/mL |
|
|
Inhibition% |
11 |
93 |
77 |
41 |
32 |
83 |
|
Sporulation% |
89 |
7 |
23 |
59 |
68 |
17 |
Materials and methods
- Shouldn't this be section 4? Check it out.
Done. Checked it out and amended
- Also pay attention to how they reference previous work. For example, in line 328. You should check the guidelines for the correct way to do this.
Done
- Line 331: why 72h? Was it long enough? Or were they based on some literature?
72 hours was about enough time for the leaves to dry completely and prepare for grinding.
- Why did you use methanol as an extraction solvent? The associated toxic effects are more than known.
Methanol is the most commonly used extraction solvent due to its high polarity which could produce high extraction yields. According to most studies extracted with methanol.
- Line 333: I don't understand. Do you percolate for 24 hours, leave the extract for 24 hours in storage, and then evaporate? Confusing. Why do you do that?
- procera Leaves extract was prepared according to the method described by Manikandan et al. (2008) with some modification
- The data on the equipment used should be complete, including information on the molecule, city, and country of production.
- Sections 2.3.1. and 2.3.2. are not being repetitive?
Recurrence has been removed
- Did you not use any commercial standards for the characterization of the compounds?
A commercial reference drug (amprolium), was used as a standard to compare compounds that are biologically active in the plant
- Was the humidity in the in vitro assays obtained through any equipment? You have to be more specific.
(ETI 6102 Therma-Hygrometer- Indian)
The 6102 thermal-hygrometer
- Why did you choose to do a one-way ANOVA?
We use a one-way ANOVA because we have data on one category independent variable and one quantitative dependent variable.
References
You should choose to reference more recent literature, preferably from the last decade.
Done.
Old references have been replaced with modern references, from 41 -46
- Perumal, G.M.; Prabhu, K.; Janaki, R.M.; Kalaivannan, J.; Kavimani, M. The GC MS Analysis of Ethyl Acetate Extract Of ‘Flueggea Leucopyrus. NVEO-NATURAL VOLATILES & ESSENTIAL OILS Journal| NVEO 2021, 4035-4040.
- Sakthivel, K.; Palani, S. Santhosh kalash R, Devi K, Senthil kumar B. Phytoconstituents analysis by GC-MS cardioprotective and anti oxidant activity of Buchania axillaris against doxorubicin induced cardiotoxicity in albino rats. International Journal of Pharmaceutical Studies and Research 2010, 1, 34-48.
- Bassaganya-Riera, J.; Hontecillas, R.; Beitz, D. Colonic anti-inflammatory mechanisms of conjugated linoleic acid. Clinical Nutrition 2002, 21, 451-459.
- Giulitti, F.; Petrungaro, S.; Mandatori, S.; Tomaipitinca, L.; De Franchis, V.; D'Amore, A.; Filippini, A.; Gaudio, E.; Ziparo, E.; Giampietri, C. Anti-tumor effect of oleic acid in hepatocellular carcinoma cell lines via autophagy reduction. Frontiers in Cell and Developmental Biology 2021, 9, 629182.
- Pegoraro, N.S.; Camponogara, C.; Cruz, L.; Oliveira, S.M. Oleic acid exhibits an expressive anti-inflammatory effect in croton oil-induced irritant contact dermatitis without the occurrence of toxicological effects in mice. Journal of Ethnopharmacology 2021, 267, 113486.
- Kalaivani, C.; Sathish, S.S.; Janakiraman, N.; Johnson, M. GC-MS studies on Andrographis paniculata (Burm. f.) Wall. Ex Nees—a medicinally important plant. Int J Med Arom Plants 2012, 2, 69-74.
I consider that the manuscript doesn't fit the scope of Molecules and doesn't have the scientific quality to be considered for publication.

Reviewer 2 Report
Combining the first and second points in the author's response, the author determined the highest abundance of phytochemical compounds in the extract of deer horn and cow leaves through GC-MSs: oleic acid=47.39%, linoleic acid=18.44%, 7-Ethyl-4-decent-6-one=10.7%, n-hexadecanoic acid=9.39%. These compounds clearly do not have anti coccidian effects. So what substances play a major role? What is the specific ratio of compounds and what is the ratio? This is the key issue that needs to be addressed in this paper.
Table 2 should not appear in this research based manuscript. Unless this manuscript is a review.
Author Response
Dear Reviewer
Thank you for accepting the review in evaluating my manuscript entitled (In Vitro: The Effects of the Anticoccidial Activities of Calotropis procera Leaf Extracts on Eimeria Stiedae Oocysts Isolated of Rabbits).
I answered all the questions and mandatory requirements in detail step by step marked in light blue
I answered all the questions and mandatory requirements in detail, step by step, marked in light blue.
Comments and Suggestions for Authors
Combining the first and second points in the author's response, the author determined the highest abundance of phytochemical compounds in the extract of deer horn and cow leaves through GC-MSs: oleic acid=47.39%, linoleic acid=18.44%, 7-Ethyl-4-decent-6-one=10.7%, n-hexadecanoic acid=9.39%. These compounds clearly do not have anti coccidian effects. So what substances play a major role? What is the specific ratio of compounds and what is the ratio? This is the key issue that needs to be addressed in this paper.
It is clear from previous studies that these compounds have an effect on microbes, infections and parasites, and it has been proven through the table 3, each compound and its effect and the reference based on it
|
Compound name |
Compound Nature |
Function |
Reference |
|
1-Amino-2,6-dimethylpiperidine |
Nature |
Cardiotoxicity Antioxidants Anti-inflammatory arthritis, and muscle and joint pain Antimicrobial |
[41] |
|
4H-Pyran-4-one, 2,3-dihydro-3,5-dihydroxy-6-methyl- |
Alkaloid |
Used for gastrointestinal disorders, gonorrhea, Menorrhagia. antifungal, anti-cancer, sclerosis, and warts |
[41] |
|
7-Ethyl-4-decen-6-one |
Flavonoid |
Antimalarial activity Antifungal Antibacterial Antidiarrhoeal activity |
[38] |
|
β-D-Glucopyranose, 1,6-anhydro |
Sugar moiety |
Antiviral, Antifungal Anthelmintic activity Antidiarrhoeal activity Anticonvulsant activity |
[38] |
|
n-Hexadecanoic acid |
Palmitic acid |
Antibacterial, Antifungal Antibiofilms Antioxidant and anticancer |
[42] |
|
Linoleic acid |
Linoleic acid |
Anti-bacterial Anti-tumor, Ant-virus, anti-inflammatory |
[43] |
|
Oleic Acid |
Palmitic acid |
Antibacterial, Antifungal Antibiofilms Antioxidant and anticancer |
[44] |
|
Octadecanoic acid |
Linoleic acid |
Antibacterial, Antifungal Antibiofilms Antioxidant and anticancer |
[45] |
|
9,12-Octadecadienoyl chloride, (Z, Z)- |
solely chloride |
Antisecretory Antispermigenic, Antitonsilitic, Antitubercular, Choleretic, Contraceptivey |
[46] |
Table 2 should not appear in this research-based manuscript. Unless this manuscript is a review.
Table 2 shows the results of the analysis of plant leaves (GCMS) consisting of 9 chemical compounds expected to be biologically active. The values for each compound appeared in figure 2, and the activities of these compounds were confirmed in Table 2 according to references.
